# A Rolling Bearing Fault Feature Extraction Algorithm Based on IPOA-VMD and MOMEDA

**DOI:** 10.3390/s23208620

**Published:** 2023-10-21

**Authors:** Kang Yi, Changxin Cai, Wentao Tang, Xin Dai, Fulin Wang, Fangqing Wen

**Affiliations:** 1School of Electronic Information, Yangtze University, Jingzhou 434023, China; 2021720695@yangtzeu.edu.cn (K.Y.); caicx@yangtzeu.edu.cn (C.C.); 2Hubei Key Laboratory of Drilling and Production Engineering for Oil and Gas, Wuhan 430100, China; 3School of Electronics and Information Engineering, Jingchu University of Technology, Jingmen 448000, China; 4Electronic and Communication Institute, China Three Gorges University, Yichang 443002, China; wenfangqing@ctgu.edu.cn; 5Institute of Vehicle Information Control and Network Technology, Hubei University of Automotive Technology, Shiyan 442002, China

**Keywords:** vibration sensor, bearing fault, pelican optimization algorithm, variational modal decomposition, Teager energy operator

## Abstract

Since the rolling bearing fault signal captured by a vibration sensor contains a large amount of background noise, fault features cannot be accurately extracted. To address this problem, a rolling bearing fault feature extraction algorithm based on improved pelican optimization algorithm (IPOA)–variable modal decomposition (VMD) and multipoint optimal minimum entropy deconvolution adjustment (MOMEDA) methods is proposed. Firstly, the pelican optimization algorithm (POA) was improved using a reverse learning strategy for dimensional-by-dimensional lens imaging and circle mapping, and the optimization performance of IPOA was verified. Secondly, the kurtosis-square envelope Gini coefficient criterion was used to select the optimal modal components from the decomposed components of the signal, and MOMEDA was used to process the optimal modal components in order to obtain the optimal deconvolution signal. Finally, the Teager energy operator (TEO) was employed to demodulate and analyze the optimally deconvoluted signal in order to enhance the transient shock component of the original fault signal. The effectiveness of the proposed method was verified using simulated and actual signals. The results showed that the proposed method can accurately extract failure characteristics in the presence of strong background noise interference.

## 1. Introduction

Rolling bearings play a crucial role in rotating mechanical equipment. Their health status directly affects the operational performance of mechanical equipment and even determines whether production processes can be performed [1]. Rolling bearing fault diagnosis mainly contains two aspects. On the one hand, the original signal is preprocessed, and the time–frequency domain’s fault features are extracted via signal processing methods. On the other hand, the extracted time–frequency domain features are used for fault identification via deep learning methods [2]. However, due to the harsh operating environment of rolling bearings and the mutual friction of the transmission system, the vibration signal contains large amounts of ambient noise, and the fault characteristics are frequently drowned out and can be challenging to discern. Therefore, the effective extraction of fault characteristics of rolling bearings in the presence of strong background noise interference can greatly improve the training efficiency and recognition accuracy of deep learning fault diagnosis algorithms, which is of immense practical significance for the normal operation of mechanical equipment [3,4].

Rolling bearing fault signals exhibit nonlinear and non-smooth characteristics, limiting the effectiveness of traditional signal processing methods for noise reduction analysis [5,6]. As such, advanced signal processing methods are required to filter out noise components from the original vibration signal [7,8]. Commonly used methods for vibration signal denoising include wavelet transform (WT), empirical mode decomposition (EMD), and ensemble EMD (EEMD) [9]. WT can characterize local signals but lacks self-adaptability and requires the manual selection of wavelet basis functions [10]. EMD decomposes the signal into a series of intrinsic mode components (IMFs) that characterize the signal; however, it is susceptible to modal aliasing and endpoint effects during the signal decomposition process [11,12]. EEMD distinguishes high- and low-frequency signals by using a mean square error criterion but suffers from long decomposition times and signal residue [13]. Although the aforementioned methods achieve noise reduction in rolling bearing fault signals to a certain extent, they fail to eliminate modal aliasing and boundary effects encountered in the signal decomposition process. Dragomiretskiy et al. [14] proposed a variational modal decomposition (VMD) algorithm to achieve adaptive signal decomposition by solving the optimal solution of variational modes. This method effectively avoids boundary effects and modal aliasing in the main frequency component, and it is more conducive to the processing of complex signals. However, the decomposition effect of the VMD algorithm is mainly determined by the penalty factor and the number of modal layers, and the improper selection of parameters can result in over-decomposition and spurious components of the signal [15,16]. Therefore, to ensure accurate parameter settings in VMD and avoid subjective errors, it is particularly important to adaptively select the optimal core parameters of VMD for subsequent feature extraction. Inspired by the application of swarm intelligence algorithms in the field of parameter optimization, this paper utilizes metaheuristic algorithms to adaptively determine the core parameters of VMD.

In recent years, scholars have applied swarm intelligence optimization algorithms to signal processing, which can not only adaptively establish the core parameters of signal processing algorithms but also improve the signal processing performance of the corresponding algorithms. Luo et al. [17] proposed an improved differential search (DS) optimization algorithm for the adaptive optimization of the core parameters of VMD and combined it with stochastic resonance theory to extract fault features. Zhang et al. [18] combined the grasshopper optimization algorithm (GOA) and the maximum weighted kurtosis index criterion to optimize the core parameters of VMD and extracted rolling bearing fault features. Yang et al. [19] used the marine predator optimization algorithm (MPA) to adaptively obtain the optimal parameters of the VMD and combined it with the fully variational denoising maximum second-order cyclic steady-state blind convolution (TVD-CYCBD) model for fault feature extraction. Ding et al. [20] proposed a VMD parameter optimization algorithm based on gene mutation particle swarm optimization (GMPSO). This method used GMPSO to obtain the optimal parameter combination of the VMD algorithm and then performed envelope spectrum analyses on the optimal modal components, finally extracting the fault features. Wang et al. [21] used an Archimedean optimization algorithm (AOA) to search for the optimal number of decomposition layers and penalty factor of the VMD and to find the IMF components that are the most sensitive to fault features. Although the above swarm intelligence algorithm can optimize the parameters of VMD to some extent, they are prone to fall into local optimization at the later stages of the iteration. Mei et al. [22] improved the pelican optimization algorithm (POA) using chaotic mapping to optimize the random forest model. However, this method only initialized the pelican population, which did not significantly improve the algorithm’s optimization seeking ability. Therefore, in order to improve the convergence speed and accuracy of POA and enhance its ability with respect to global optimization searching, this paper proposes a new improvement strategy to adaptively adjust the position of pelican individuals in POA via circle mapping and reverse learning strategies for dimensional-by-dimensional lens imaging.

After the preliminary noise reduction in the original signal using the signal decomposition algorithm, methods for extracting the periodic shock signal from the signal components are also one of the key steps in fault feature extraction. To address this problem, Endo et al. [23] proposed a minimum entropy deconvolution (MED) algorithm. However, this method can only obtain a single pulse signal, while the rolling bearing fault signal is generally a periodic pulse signal. McDonald et al. [24] proposed a maximum correlation kurtosis deconvolution (MCKD) algorithm by combining correlation kurtosis and the deconvolution algorithm. However, the model of the algorithm is complex, and the accuracy of the model is determined by multiple parameters. McDonald et al. [25] proposed a multipoint optimal minimum entropy deconvolution-adjusted (MOMEDA) algorithm and employed it for the fault diagnosis of rolling bearings [26,27,28]. MOMEDA eliminates the need for preset failure cycles, and the model is highly generalizable. MOMEDA overcomes the limitations of MED and MCKD, which require constant iterations to obtain the optimal filter, and it can effectively enhance and extract the periodic shock components in vibration signals.

To effectively and precisely extract failure characteristics in the presence of strong background noise, we propose applying the IPOA-VMD and MOMEDA algorithms to the fault detection of rolling bearings. Firstly, the POA is improved using circle mapping and a reverse learning strategy for dimensional-by-dimensional lens imaging, and the core parameters of VMD are adaptively optimized using IPOA. Secondly, the kurtosis-square envelope Gini coefficient (*K-SEGI*) criterion is used to select the optimal modal components from the decomposed components of the signal, and MOMEDA is used to further denoise the optimal modal components to obtain the optimal deconvolution signal. Finally, the Teager energy operator (TEO) is employed in demodulation, and the optimally deconvoluted signal is analyzed, achieving the accurate extraction of fault features. We verified the effectiveness of the proposed method for failure characteristic extraction in the presence of strong background noise interference using simulated and actual signals.

## 2. Variational Mode Decomposition (VMD)

The constrained variational model constructed by the VMD is described as follows [29]:(1)$$\left\{\begin{array}{l} \underset{\left\{u_{k}\right\},\left\{\omega_{k}\right\}}{\min}\left\{\sum_{k=1}^{K}{\left\|\alpha_{t}\left[\left(\left(\delta (t)+\frac{j}{\pi t})*u_{k}(t\right)\right)\right]e^{-j\omega_{k}t}\;\right\|}_{2}^{2}\right\}\\ s.t.\sum_{k=1}^{K}u_{k}=f \end{array}\right\}$$
where $\left\{u_{k}\right\}$ refers to each IMF component that is derived from the VMD decomposition, and *f* denotes the original signal. The $\left\{\omega_{k}\right\}$ represents the center frequency associated with each IMF component, δ(t) represents the unit pulse function, and * represents convolutional operations.

Using an augmented Lagrange function, we can transform a restricted variational problem into an unrestricted variational problem. The expression is as follows:(2)$$\begin{array}{r} L\left(\left\{u_{k}\right\},\left\{\omega_{k}\right\},\lambda \right)=\alpha \sum_{k=1}^{K}{\left\|\alpha_{t}\left[\left(\delta (t)+\frac{j}{\pi t})*u_{k}(t\right)\right]e^{-j\omega_{k}t}\right\|}_{2}^{2}+{\left\|f(t)-\sum_{k=1}^{K}u_{k}(t)\;\right\|}_{2}^{2}\\ +\langle \lambda (t),f(t)-\sum_{k=1}^{K}u_{k}(t)\rangle \end{array}$$
where *λ* is the Lagrange multiplier, and *α* is the punishment factor.

The alternating direction method of multipliers (ADMMs) is employed to address the problem of solving the unconstrained variational model. The update formulas for $\left\{\omega_{k}\right\}$, $\left\{u_{k}\right\}$, and λ, as well as the iteration termination conditions, are as follows:(3)$$\omega_{k}{}^{n+1}=\frac{\int_{0}^{\infty }{\omega |\;\hat{u}}_{k}{\left(\omega )\right|}^{2}d\omega }{\int_{0}^{\infty }{|\;\hat{u}}_{k}{\left(\omega )\right|}^{2}d\omega }$$
(4)$${\;\hat{u}}_{{}_{k}}^{\;n+1}(\omega )=\frac{\;\hat{f}\;(\omega )-\sum_{i\neq k}{\;\hat{u}}_{i}(\omega )+\frac{\;\hat{\lambda }\;(\omega )}{2}}{1+2\alpha (\omega -\omega_{k}{)}^{2}}$$
(5)$${\;\hat{\lambda }}^{n+1}{(\omega )=\hat{\lambda }}^{n}\left(\omega )+\tau (\;\hat{f}\;(\omega \right)-\sum_{k=1}^{k}{\;\hat{u}}_{k}{}^{n+1}\left(\omega \right))$$
(6)$$\sum_{k=1}^{k}\left({{\left\|{\;\hat{u}}_{k}{}^{n+1}-{\;\hat{u}}_{k}{}^{n}\right\|}_{2}^{2}/\left\|{\;\hat{u}}_{k}{}^{n}\right\|}_{2}^{2}\right)\;<\;\varepsilon$$
where ${\hat{u}}_{{}_{k}}^{\;n+1}$, ${\hat{\lambda }}^{n+1}$, and $\hat{f}$ are, respectively, denote the Fourier transform corresponding to $u_{{}_{k}}^{\;n+1}$, $\lambda^{n+1}$, and *f*; ε is the convergence accuracy. Cyclic updating is performed using the aforementioned steps until the termination condition is met, finally yielding the *K* IMF components.

## 3. IPOA-VMD and MOMEDA Fault Feature Extraction Algorithms

### 3.1. Improved Pelican Optimization Algorithm (IPOA)

POA achieves optimal searches by simulating the hunting process of pelican populations in terrestrial organisms [30]. Compared to commonly used optimization algorithms [31,32,33], POA has advantages, such as fast convergence speed, resilience to local optima, and strong approximation ability to find optimal solutions [34]. POA comprises two main phases: the exploration phase and the development phase. However, the problem of decreasing population diversity occurs at the later stage of iterations. Therefore, in this paper, we proposed the use of circle mapping and reverse learning strategies for dimensional-by-dimensional lens imaging to enhance population diversity. First, circle mapping is employed to initialize the positions of the pelican population so that each pelican is evenly distributed across the entire search space. Next, the position of each pelican is optimized using the reverse learning strategy for dimensional-by-dimensional lens imaging to enhance the diversity of the pelican population during the later iteration process of the POA in order to improve the convergence rate of the algorithm and reduce the risk of trapping in local optimality.

#### 3.1.1. Circle Mapping Strategy

Chaotic mapping exhibits properties such as randomness, ergodicity, and regularity, which can be utilized to improve the optimization performance of POA. Commonly used chaotic mappings in the field of optimization include logistic mappings, tent mappings, and circle mappings [35]. As observed in the distribution of the 1500 sequence values generated using four different methods, as shown in Figure 1, circle mapping generated a more uniform and stable distribution of chaotic sequence values between 0 and 1 compared to logistic mapping, tent mapping, and ordinary random numbers. Therefore, in this study, circle mapping was used to initialize the positions of the pelican population. Circle mapping can be defined as follows:(7)$$num_{i+1}=mod\left(num_{i}+0.2-\left(\frac{0.5}{2\pi }\right)sin(2\pi num_{i}),1\right)$$
where $num_{i}$ is the *i*-th chaotic sequence number, and mod is the remainder operation.

The expression for IPOA population initialization is as follows:(8)$$X_{ij}=l_{j}+(u_{j}-l_{j})\times num_{ij}$$
where $num_{ij}$ is the value of the chaotic sequence generated via circle mapping on the dimension of the corresponding pelican individual.

#### 3.1.2. Reverse Learning Strategy for Dimensional-by-Dimensional Lens Imaging

Two methods are mainly used to solve the local optimum problem: (1) maintaining the current optimal position and expanding the search area and (2) abandoning the current optimal position and searching in a new area. In this study, the first method was used, inspired by lens imaging [36], and the reverse learning strategy for dimensional-by-dimensional lens imaging was used to facilitate the POA in escaping from the local optimal region. Figure 2 illustrates the schematic of the reverse learning strategy for lens imaging [36].

For a space with a search range of [*a_j_*, *b_j_*] for feasible solutions, the position of the optimal individual $X_{best}^{j}$ in the *j*-th dimension represents the projection of an object p with a height of *h* on the x-axis. A convex lens is placed at base point *o*. An object *p* creates an inverted real image *p’* with height *h’* on the other side of the convex lens. At this point, the projection of *p’* on the x-axis is represented as ${X^{\prime }}_{best}^{j}$. The following expression is based on the principle of lens imaging:(9)$$\frac{(a_{j}+b_{j})/2-X_{best}^{j}}{{X^{\prime }}_{best}^{j}-(a_{j}+b_{j})/2}=\frac{h}{h^{\prime }}$$

Let $h/h^{\prime }=n$. Accordingly, Equation (9) can be transformed as follows:(10)$${X^{\prime }}_{best}^{j}=\frac{\left(a_{j}+b_{j}\right)}{2}+\frac{\left(a_{j}+b_{j}\right)}{2n}-\frac{X_{best}^{j}}{n}$$

When *n* = 1, we obtain the following:(11)$${X^{\prime }}_{best}^{j}=(a_{j}+b_{j})-X_{best}^{j}$$

As observed in Equation (11), when *n* = 1, a fixed reverse solution is obtained. Thus, the dynamically varying inverse solution is obtained by adjusting the value of *n*. First, the optimal individual position of the pelican is updated using the proposed optimization strategy, mapping the positions of each dimension into space to obtain the reverse position. Next, the previous position’s fitness value is compared with the fitness value after reverse learning. If the fitness value after reverse learning is better than that of the previous position, the reverse position is selected to replace the previous position. Otherwise, the original position is retained for the next generation. The reverse learning strategy for dimensional-by-dimensional lens imaging not only avoids interference between different dimensions but also expands the search range of the algorithm.

#### 3.1.3. IPOA Algorithm Performance Experiment

To verify the optimization performance of IPOA, the swarm intelligence algorithms in the literature [17,18,19,20,21] were compared with IPOA. Multiple benchmark functions were used for testing. The benchmark test function parameters are presented in Table 1. During the testing process, for all the algorithms, the limit for the number of iterations was 100, the initial population was set as 30, and the search space dimension of the population was 30. To avoid the randomness of the experiment, each algorithm was run independently 30 times for each benchmark function. The results of 30 experiments were counted, and the average value Avg and the total number of times to reach the target optimal value MR (the target optimum value was set to 10^−6^) were calculated. The smaller the value of Avg, the stronger the ability of the optimization algorithm to approximate the optimal solution and the greater the probability of reaching the global optimum. The larger the value of MR, the higher the optimization accuracy; moreover, convergence speeds become faster, and the optimization algorithm exhibits stronger reliability. The iterative curves of different optimization algorithms under single mode and multimodal benchmark test functions are displayed in Figure 3 and Figure 4, respectively. The experimental results of different optimization algorithms are shown in Table 2.

The above chart clearly shows the test results of different optimization algorithms. As observed in Figure 3 and Figure 4 and Table 2, IPOA converged stably to the optimal solution for single-mode benchmark test functions *F*_1_ and *F*_2_. It exhibited a better convergence rate, optimization accuracy, and stability compared to the optimization algorithm proposed in the literature [17,18,19,20,21]. For multimodal benchmark test functions *F*_3_ and *F*_4_, IPOA also converged stably to the optimal solution. By employing a reverse learning strategy to avoid falling into local optima in the later stages, the optimization effect was significantly improved. In summary, IPOA exhibits strong stability and robustness.

### 3.2. Optimization of IPOA-VMD Parameters

#### 3.2.1. Envelope Spectral Entropy (ESE)

To optimize the core parameters of VMD in IPOA, an appropriate fitness function must be constructed. We introduced the concept of entropy to ensure rational parameter selection. Entropy is an indicator used to reflect the sparsity characteristics of a signal. A smaller entropy value indicates that the sequence contains more meaningful information and is smoother [37]. Among information entropy metrics, envelope spectral entropy (ESE) has the characteristics of simple calculation and fewer parameter inputs [38]. Therefore, in this paper, ESE was employed as the fitness function for the IPOA-VMD optimization model. The expression for ESE is as follows:(12)$$\left\{\begin{array}{l} E_{p}=-\sum_{i=1}^{N}\varepsilon (i)lg\varepsilon (i)\\ \varepsilon (i)=\frac{a(i)}{\sum_{i=1}^{N}a(i)} \end{array}\right.$$
where *a*(*i*) is the envelope signal of the original signal after Hilbert demodulation, the length of the signal can be denoted as *N*, and *ε*(*i*) is the normalized form of *a*(*i*).

#### 3.2.2. IPOA-VMD Optimization Flow

The flow of IPOA for optimizing the core parameters of VMD is illustrated in Figure 5. The concrete steps can be outlined as follows:

Step 1: The maximum number of iterations, spatial dimensions, population size, decomposition layers, and the penalty factor for IPOA are set, and the circle mapping strategy is used to initialize the population’s position.

Step 2: The VMD decomposition of the original signal yields several IMF components. The ESE value is calculated for each IMF component, and the fitness function for global exploration is determined by selecting the component with the lowest ESE value.

Step 3: After each round of iteration, the ESE value corresponding to each set of parameter combinations is calculated and compared with the current ESE value. If it is less than the current ESE value, the current ESE value will be renewed.

Step 4: Whether the iteration’s stop condition has been reached is determined. If the maximum number of iterations is not reached, let *t* = *t +* 1. In addition, the pelican population position updated in the previous iteration is used as the initial population position in the next round. Steps 1–5 are repeated until the iteration condition is reached.

Step 5: When the loop iteration ends, the optimal parameter combination will be outputted.

**Figure 5 sensors-23-08620-f005:**
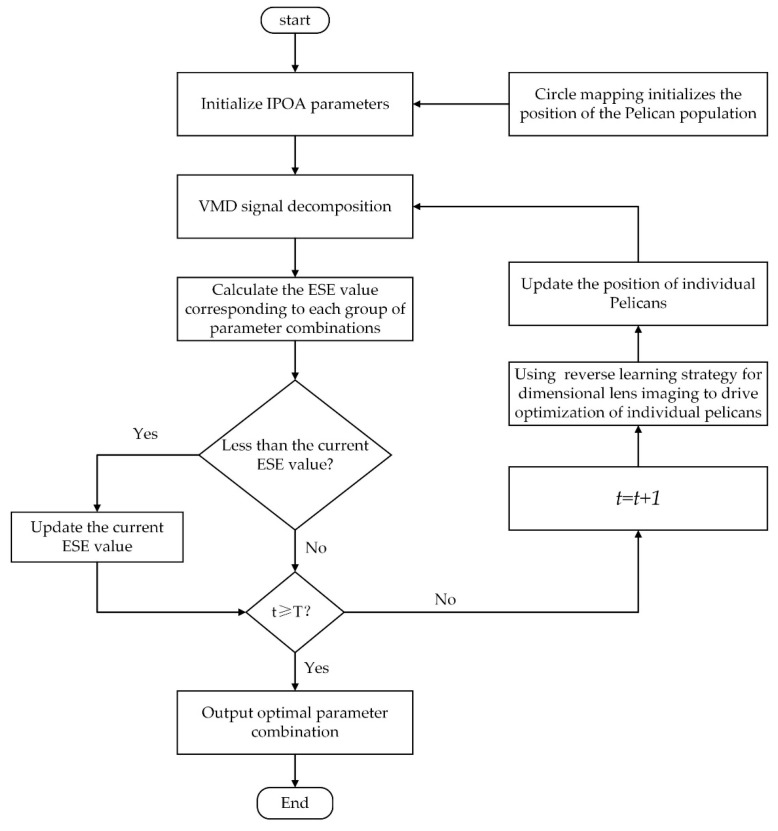
Flowchart of the IPOA for the optimization of VMD parameters.

### 3.3. Component Screening and MOMEDA Algorithm

#### 3.3.1. Component Screening

Selecting the optimal component from multiple IMF components is important for accurately extracting fault features. Metrics such as kurtosis, correlation coefficient, and sample entropy are generally used to screen valid components [18]. However, a single screening metric is susceptible to noise interference, resulting in the misidentification of valid components [39].

The Gini coefficient is an index used for measuring the sparsity of a sequence [40], and it has been used in the fault diagnosis of rolling bearings due to its high stability against noise disturbances [41,42]. To better screen the effective components, in this study, the squared envelope Gini coefficient (*SEGI*) was used. The value of *SEGI* is between 0 and 1; the closer it is to 1, the better the balance of the sequence. The expression for *SEGI* is as follows:(13)$$SEGI=1-2\sum_{n=1}^{N}\frac{SE_{\left(n\right)}}{∥SE{∥}_{1}}(\frac{N-n+0.5}{N})$$
(14)$$SE=[SE_{\left(1\right)},SE_{\left(2\right)},\ldots ,SE_{\left(i\right)},\ldots SE_{\left(N\right)}]\;\;(SE_{\left(1\right)}\leq SE_{\left(2\right)}\leq SE_{\left(i\right)}\leq SE_{\left(N\right)})$$
where *SE* represents the squared envelope of the original signal, and $∥SE{∥}_{1}$ is the L1 paradigm of the *SE*.

Kurtosis is sensitive to shock signals and is a good indicator for detecting periodic shocks. The expression for kurtosis can be described as follows:(15)$$K=\frac{1}{N}\sum_{i=1}^{N}{\left(\frac{x-x_{i}}{\sigma }\right)}^{4}$$
where *x* is the expected value, the value of *N* represents the total number of signal points, and σ denotes the standard deviation.

To utilize the advantages of these two indicators, we proposed a screening criterion called *K-SEGI*. Due to the different dimensions and value ranges of these two indicators, it is necessary to standardize their amplitudes. First, the amplitude is normalized. Next, the normalized amplitude is exponentially increased based on a base of 2, and the resulting value is used as the final amplitude. The calculation formula for the *K-SEGI* screening criterion is as follows:(16)$$K-SEGI={\left(K^{\prime }\cdot SEGI^{\prime }\right)}_{max}$$
where *K’* and *SEGI’* are the amplitudes of the two indicators after normalization.

#### 3.3.2. MOMEDA Algorithm

MOMEDA is a weak signal enhancement method and is a non-iterative deconvolution process for obtaining optimal finite impulse response (FIR) filters [25]. Assuming that the original vibration signal is *x*, the following expression holds:(17)x=h*y+e
where *y* is the periodic shock signal, *h* is the transfer function, *e* is the ambient noise, and * denotes the convolution operation.

MOMEDA recovers the periodic impulse signal y by searching for the optimal FIR filter. The process of solving the optimal filter can be translated into finding the maximum value of the multipoint D-paradigm number (*MDN*):(18)$$MDN(y,t)=\frac{1}{\left\|t\right\|}\frac{t^{T}y}{\left\|y\right\|}$$
where fault period *T* can be defined as the ratio between the sampling frequency and the eigenfrequency of the fault, and *t* represents the target vector that signifies the position and weight of the deconvolution target shock signal. When *MDN* reaches the maximum value, the following expression holds:(19)$$max\;MDN(y,t)=\underset{f}{max}\frac{t^{T}y}{\left\|y\right\|}$$

By taking the derivative of Equation (19) and setting it to 0, the optimal filter for MOMEDA can be obtained as follows:(20)$$\frac{d}{df}\left(\frac{t^{T}y}{\left\|y\right\|}\right)=\frac{d}{df}\left(\frac{t_{1}y_{1}}{\left\|y\right\|}\right)+\cdots +\frac{d}{df}\left(\frac{t_{N-L}y_{N-L}}{\left\|y\right\|}\right)$$

In addition,
(21)$$\frac{d}{df}\left(\frac{t_{i}y_{i}}{\left\|y\right\|}\right)=t_{i}M_{i}{\left\|y\right\|}^{-1}-{\left\|y\right\|}^{-3}t_{i}y_{i}X_{0}y\;,\;M_{i}=\left[\begin{array}{c} x_{i+L-1}\\ \vdots \\ x_{i} \end{array}\right]$$

Substituting Equation (21) into Equation (20) yields
(22)$$\frac{d}{df}\left(\frac{t^{T}y}{\left\|y\right\|}\right)=(t_{1}M_{1}+t_{2}M_{2}+\cdots +t_{N-L}M_{N-L}){\left\|y\right\|}^{-1}-{\left\|y\right\|}^{-3}t^{T}yX_{0}y$$

Let $t_{1}M_{1}+t_{2}M_{2}+\cdots +t_{N-L}M_{N-L}=X_{0}t$, and collation is carried out to obtain
(23)$$(X_{0}t){\left\|y\right\|}^{-1}-{\left\|y\right\|}^{-3}t^{T}yX_{0}y=0$$

Because $y=X_{0}{}^{T}f$ and assuming ${\left(X_{0}X_{0}{}^{T}\right)}^{-1}$ exists, we obtain
(24)$$\frac{t^{T}y}{{\left\|y\right\|}^{2}}={\left(X_{0}X_{0}{}^{T}\right)}^{-1}X_{0}t$$

In summary, the optimal filter of MOMEDA can be expressed as follows:(25)$$f={\left(X_{0}X_{0}{}^{T}\right)}^{-1}X_{0}t$$

Features extraction using only MOMEDA is less effective due to the interference of strong background noise. To address this problem, TEO [43] is used to demodulate and analyze the optimal deconvolution signal obtained after MOMEDA processing to further suppress the interference of noise in order to accurately extract fault characteristics.

### 3.4. Fault Feature Extraction Method Process

The method for extracting fault features is outlined in Figure 6. The method consists of the following steps:

Step 1: VMD core parameters are optimized using IPOA, and the original fault signal is adaptively decomposed.

Step 2: The *K-SEGI* value for each IMF component after signal decomposition is calculated.

Step 3: The modal component with the highest *K-SEG*I value is chosen as the optimal IMF component.

Step 4: The optimal modal component is further denoised using MOMEDA to obtain the optimal deconvolution signal.

Step 5: The optimal deconvolution signal is analyzed using TEO demodulation to extract the bearing’s fault features.

**Figure 6 sensors-23-08620-f006:**
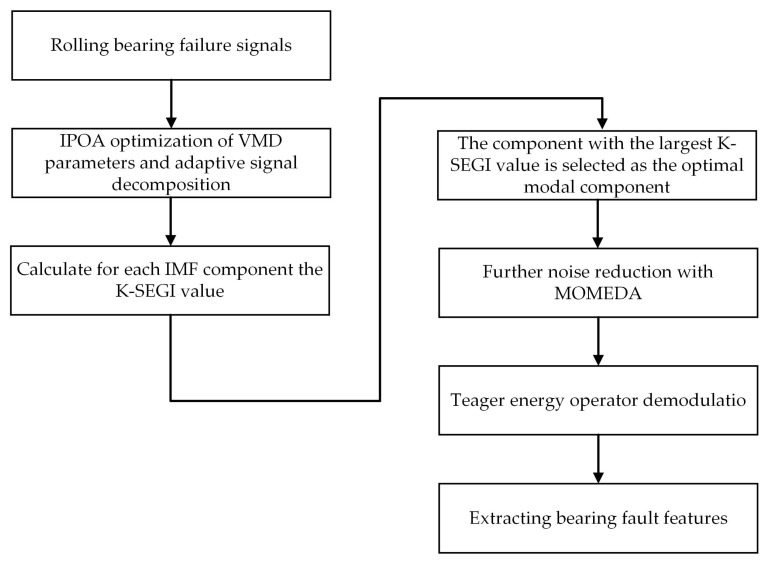
Fault feature extraction process.

## 4. Experimental Study

### 4.1. Simulated Signal Analysis

#### 4.1.1. Establishment of Simulation Signals

We construct a generated set of periodic pulse signals, and Gaussian white noise is added. The simulated signal can be formulated in the following manner:(26)$$\left\{\begin{array}{l} y(t)=h(t)+n(t)\\ h(t)=\sum_{i}A_{i}x(t-iT)\\ x(t)=e^{-\alpha t}sin(2\pi f_{n}t)\\ A_{i}=1+A_{0}sin(2\pi f_{r}t) \end{array}\right.$$
where *n*(*t*) represents the Gaussian white noise signal, *y*(*t*) denotes the simulated signal for bearing faults, *h*(*t*) represents the periodic shock signal, *t* represents the sampling time, *T* represents the repetition period, *A*_0_ represents the displacement constant, *α* represents the attenuation constant, *f_r_* is the rotation frequency of the bearing, and *f_n_* is the intrinsic frequency.

For *A*_0_ = 0.01, *α* = 700, and *t* = 0.2 s, the number of sampling points is 2400, intrinsic frequency is *f_n_* = 4 kHz, bearing rotation frequency is *f_r_* = 30 Hz, repetition period is *T* = 0.0067 s, characteristic frequency of the bearing’s failure is *f_i_* = 149.25 Hz, and sampling frequency is *f_s_* = 12 kHz. Figure 7 illustrates the basic characteristics of the periodic shock signal.

To create a simulation of high background noise environments, Gaussian white noise with a signal-to-noise ratio (SNR) of −15 dB was introduced to the periodic shock signal. Figure 8 displays the basic characteristics of the simulated signal. As depicted in Figure 8a, the characteristics of the periodic shock signal are heavily contaminated by Gaussian white noise signals. Furthermore, as depicted in Figure 8b, noise submerged the characteristic frequency and its octave in the simulated signal, making it impossible to extract fault features from the simulated signal.

#### 4.1.2. IPOA-VMD Signal Decomposition

The adaptive decomposition of the simulated signal was performed using IPOA-VMD. The initialization parameters for IPOA are presented in Table 3. The optimization iteration curve of IPOA-VMD is shown in Figure 9.

As illustrated in Figure 9, the ESE value decreased with the increase in the number of iterations during the optimization process, and the minimum ESE value occurred in the 15th generation, after which the minimum ESE value remained constant. When the loop iteration ended, the optimal combination of parameters was [4, 19,219]. The VMD decomposition of the simulated signal by using the optimal parameter combinations yielded four IMF components. The waveforms of these components are displayed in Figure 10a,b, both in the time and frequency domains.

As illustrated in Figure 10b, the simulated signals were distributed in different frequency bands after the adaptive decomposition of IPOA-VMD, effectively avoiding the boundary effects and modal aliasing phenomena. To accurately select the optimal modal components from each IMF component, the evaluation parameters of each IMF component were calculated and are presented in Table 4. As evident from Table 4, the *K-SEGI* value of IMF1 was the largest, indicating that this component contained more shock components and less noise content; thus, it was selected as the optimal modal component. The kurtosis index selected IMF1 as the optimal modal component, while the *SEGI* index selected IMF4 as the optimal modal component. This indicated that a single screening indicator can easily lead to the misidentification of the optimal modal component, which proved the superiority of the comprehensive screening indicator proposed in this paper.

#### 4.1.3. Fault Feature Extraction

IMF1 was further denoised using MOMEDA by setting the period of deconvolution as *T* = *f_s_*/*f_r_* = 80 and the filter length as *L* = 700. The optimal deconvolution signal time-domain waveform of IMF1 after MOMEDA processing is shown in Figure 11, and the results obtained via demodulation analysis using TEO are displayed in Figure 12. As observed in Figure 11, the periodic shock component of the simulated signal was very obvious, and the noise reduction effect was good.

The results analyzed for IMF1 using only TEO demodulation are depicted in Figure 13a, and the results processed using only MOMEDA are depicted in Figure 13b. By comparing Figure 12 and Figure 13, it can be observed that the analyses of the signal using only TEO demodulation and processing using only MOMEDA are not satisfactory, the fault characteristics are not obvious, and the peaks are not prominent. This suggested that using TEO or MOMEDA alone did not accurately extract the fault features, and the characteristic frequency is still disturbed by a small amount of noise. In contrast, the fault characteristics frequency of IMF1 obtained via demodulation analysis using MOMEDA combined with TEO were extremely obvious, and peak *f_i_* and its octave were very prominent, corresponding to *f_i_*–6*f_i_*. Therefore, the proposed method enables the precise extraction of fault characteristics.

#### 4.1.4. Signal Processing Performance of the Proposed Method

In order to prove the reliability of the method proposed in this paper, experiments were conducted by adding noise signals with an SNR of −16 dB, −17 dB, and −18 dB to the pure signal. According to the calculation formula of SNR, the noise power in the above four (including the noise signal with SNR of −15 dB) simulated signals was 31.6 times, 39.8 times, 50.1 times, and 63.1 times the signal power, respectively. Figure 14 shows the spectrum of the components of each simulated signal after IPOA-VMD decomposition, and Figure 15 shows the TEO spectrum of the optimal IMF of each simulated signal after MOMEDA processing. As observed in Figure 14, the IMFs are distributed in different frequency bands without mode aliasing.

As observed in Figure 15, the peaks of fault characteristic frequency and its octave are extremely prominent. This proved the reliability of the proposed method in this paper at different SNR levels. During the experiment, when a noise signal with an SNR of −20 dB was added to the pure signal (the noise power was 100 times the signal power), it was observed that the signal processing time was too long and always exceeded the limit value of the preset parameters. This indicated that the limits of the signal processing method proposed in this paper were about to be reached.

### 4.2. Measured Signal Analysis

The experimental data were obtained from the vibration acceleration signal collected by a university in the US on the rolling bearing fault experimental platform [44]. Figure 16 illustrates the schematic of the experimental platform. The basic parameters of the bearing are presented in Table 5 [44]. In this paper, the fault data of the inner race of the bearing at the driving end of four kinds of motors with different actual load power (0 kW, 0.735 kW, 1.47 kW, and 2.205 kW) are selected for experimental analyses. The motor load is generally measured using percentages, which are calculated as the ratio of the actual load power to the rated power of the motor multiplied by 100%. The number of sample points is 2400, the sampling frequency *f_s_*_2_ is 12 kHz, and the rated power of the motor is 1.47 kW. Table 4 lists the parameters, such as the speed, diameter, and motor load of the bearings. The motor rotation frequency *f_r_* and the characteristic frequency *f_i_* of the inner race fault were calculated as follows:(27)$$\left\{\begin{array}{l} f_{i}=0.5Z(1+\frac{d}{D}cos\beta )f_{r}\\ f_{r}=\frac{n}{60} \end{array}\right.$$
where *D* represents the pitch diameter of the bear’ng’s raceway, *Z* represents the number of balls, *d* represents the diameter of the ball, *β* represents the contact angle of the bearing, and *n* represents the motor’s speed.

Based on the parameters presented in Table 6, when the motor load was 0% (indicating that the motor was in a no-load state), the fault characteristic frequency was calculated using *f_i_*_2_ = 162.19 Hz in Equation (27), and the motor rotational frequency was calculated as *f_r_*_2_ = 29.95 Hz. To create a simulation of a high background noise environment, Gaussian white noise with a signal-to-noise ratio (SNR) of −6 dB was introduced to the actual signal. Figure 17 displays the basic characteristics of the inner ring fault signal.

The adaptive decomposition of inner-ring bearing fault signals was performed using IPOA-VMD. The initialization parameters for IPOA are presented in Table 7. The IPOA-VMD optimization search iteration curve is displayed in Figure 18.

As observed in Figure 18, the minimum ESE value occurred in the 11th generation, after which the minimum ESE value remained constant. When the loop iteration ended, the optimal combination of parameters was [3, 2536]. We applied the optimal parameter combination to perform VMD decomposition on the inner ring fault signal, and three modal components were obtained. The waveforms of these components are displayed in Figure 19a,b, both in the time and frequency domains.

The evaluation parameters of each IMF component are presented in Table 8. As evident from Table 8, the *K-SEGI* value of IMF2 was the largest. Thus, IMF2 was selected as the optimal modal component. As observed in Figure 19a, the fault shock characteristics of IMF2 were more obvious; however, a small amount of noise interference remained. Thus, MOMEDA was used for further noise reduction in IMF2 by setting the period of deconvolution as *T* = *f_s_*_2_/*f_r_*_2_ = 73.99 and filter length as *L* = 700. The optimal deconvolution signal time-domain waveform of IMF2 after MOMEDA processing is shown in Figure 20, and the results obtained via performing demodulation analysis using TEO are shown in Figure 21.

To further illustrate the effectiveness of the method proposed in this paper, the rolling bearing inner ring fault data with motor loads of 50%, 100%, and 150% were selected for experiments. According to Equation (27), when the motor load is 50%, the fault characteristic frequency *f_i_*_3_ is 159.91 Hz, and the rotation frequency *f_r_*_3_ is 29.53 Hz. When the motor load is 100%, the fault characteristic frequency *f_i_*_4_ is 157.96 Hz, and the rotation frequency *f_r_*_4_ is 29.17 Hz. When the motor load is 150%, the fault characteristic frequency *f_i_*_5_ is 156.12 Hz, and the rotation frequency *f_r_*_5_ is 28.83 Hz. According to the above fault feature extraction process, Gaussian white noise with an SNR of −6 dB was added to the inner race signals of rolling bearings under different loads, and the parameters of the decomposition algorithm were consistent with those shown in Table 7. Figure 22 displays the time-domain waveforms and TEO spectrum of each optimal IMF after MOMEDA processing.

As observed in Figure 22, when the motor load is 50%, the actual frequency of fault characteristics is 159.42 Hz. When the motor load is 100%, the actual frequency of fault characteristics is 157.52 Hz. When the motor load is 150%, the actual frequency of fault characteristics is 156.21 Hz. They are all very close to the theoretically calculated frequency, and the peak and octave frequencies are extremely prominent. This showed that the algorithm proposed in this paper can accurately extract the characteristics of the inner race fault signal under different motor load conditions. This proved the effectiveness and reliability of the proposed method.

### 4.3. Comparative Analysis of Different Feature Extraction Methods

To substantiate the excellence of the proposed method in fault feature extraction in the presence of strong background noise interference, the EEMD algorithm, fixed-parameter VMD algorithm (FP-VMD), and empirical wavelet decomposition–independent component analysis (EWT-ICA) algorithm were compared with the proposed method in this paper. The penalty factor in the FP-VMD algorithm was 20,000, and the number of decomposition layers was eight. The EWT-ICA algorithm first performed an EWT decomposition of the signal. Then, several IMF components were selected as observation signals for the ICA algorithm based on the screening index, and other IMF components were used as virtual noise channel signals for the algorithm. Finally, the obtained source signals were enveloped and demodulated to extract the fault features. To increase the credibility of the experiment, the actual signals and screening metrics selected are consistent with this paper. Figure 23 shows the optimal IMF envelope spectra obtained using the EEMD algorithm for rolling bearing inner race fault signals under different loads. Figure 24 shows the optimal IMF envelope spectrum obtained using the FP-VMD algorithm for rolling bearing inner race fault signals under different loads. Figure 25 shows the optimal source signal envelope spectrum obtained using the EWT-ICA algorithm for rolling bearing inner race fault signals under different loads.

As observed in Figure 23, Figure 24 and Figure 25, the EEMD, PF-VMD, and EWT-ICA algorithms were only able to extract the rotational frequency of the motor for the signals of the inner race of the rolling bearing under different loads, and the characteristic frequency and its multiplicative frequency were swamped by noise. In addition, the EEMD algorithm was only able to extract the rotational frequency of the motor when there was no load. but all the other characteristic frequencies are submerged by noise. In contrast, as observed in Figure 21 and Figure 22, the actual characteristic frequency of the inner race fault approached the theoretical value in an extreme manner, and the fault characteristics were very obvious, enabling the accurate extraction of the characteristic frequency and its multiplicative frequency. The above results indicated that the EEMD, FP-VMD, and EWT-ICA algorithms cannot fully extract fault features in strong background noise environments, thus fully demonstrating the superiority of the method proposed in this paper.

## 5. Conclusions

We combine the improved POA-VMD with MOMEDA-TEO to research the rolling bearing fault feature extraction method in the presence of strong background noise environments. The main conclusions of this paper are as follows:(1)The POA is improved to form the IPOA using circle mapping and reverse lens imaging learning strategies, and the IPOA is used to optimize the penalty factor and the number of modal layers in the VMD algorithm to adaptively obtain the optimal parameter combination. This method overcomes the over-decomposition of the signal and modal aliasing problems caused by the improper setting of parameters based on human experience and subjectivity.(2)Based on the advantages of *SEGI* and kurtosis screening metrics, a new optimal modal component screening metric of *K-SEGI* is proposed. It can effectively screen the components with the most periodic shocks and the most stable components while avoiding the mis-selection problem of the optimal component caused by using a single screening metric of *SEGI* and kurtosis.(3)After the preliminary noise reduction in the original signal using the IPOA-VMD decomposition algorithm, the optimal IMF component is further denoised by using MOMEDA, and the impact signal is highlighted better using TEO demodulation analyses. We conducted fault extraction experiments using simulated signals and four measured signals of inner race fault under different loads. The experimental results show that the proposed method can accurately extract the fault frequency and its octave. In addition, compared with EEMD, FP-VMD, and EWT-ICA feature extraction algorithms, the proposed method exhibits obvious advantages with respect to fault feature extraction in strong background noise environments.

## Figures and Tables

**Figure 1 sensors-23-08620-f001:**
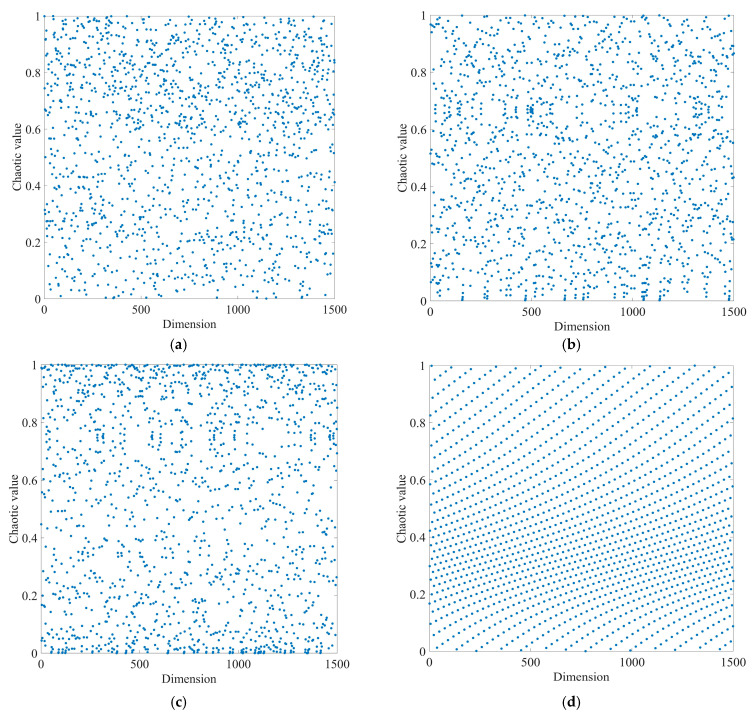
Distribution map of sequence values: (**a**) random number generation, (**b**) tent mapping generation, (**c**) logistic mapping generation, (**d**) circle mapping generation.

**Figure 2 sensors-23-08620-f002:**
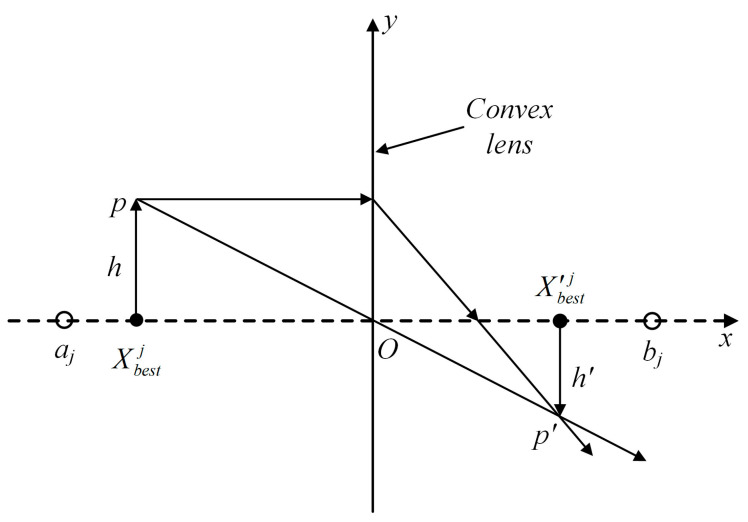
Schematic of the reverse learning strategy for lens imaging.

**Figure 3 sensors-23-08620-f003:**
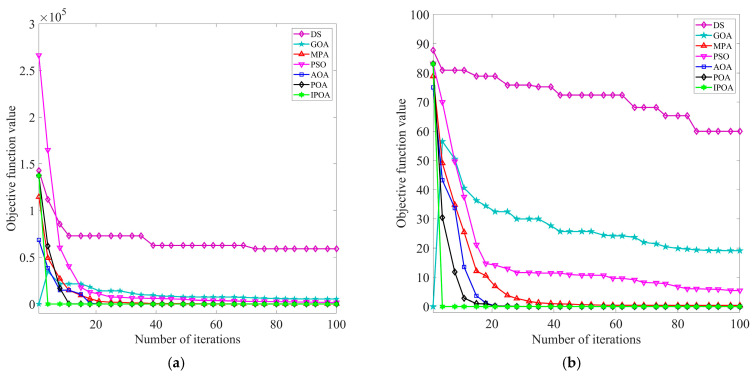
Iterative curves of different optimization algorithms under the single modal benchmark function: (**a**) iteration curve for benchmark test function *F*_1_ and (**b**) iteration curve for benchmark test function *F*_2_.

**Figure 4 sensors-23-08620-f004:**
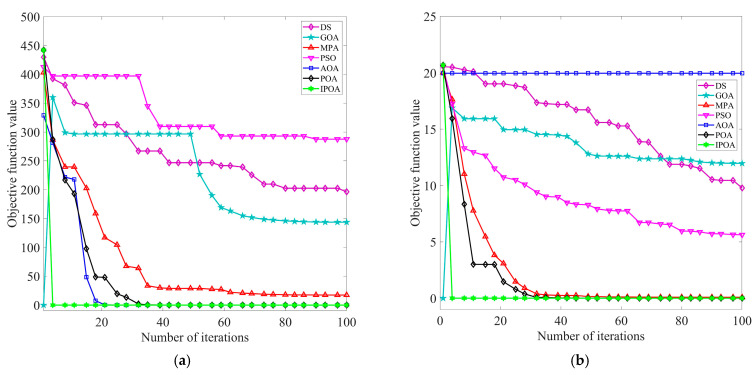
Iterative curves of different optimization algorithms under multimodal benchmark functions: (**a**) iteration curve for benchmark test function *F*_3_ and (**b**) iteration curve for benchmark test function *F*_4_.

**Figure 7 sensors-23-08620-f007:**
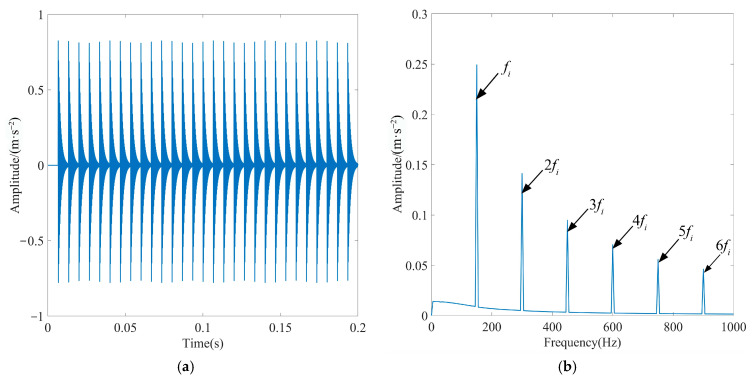
Basic characteristic diagram of the periodic shock signal: (**a**) time-domain diagram and (**b**) Hilbert envelope spectrogram.

**Figure 8 sensors-23-08620-f008:**
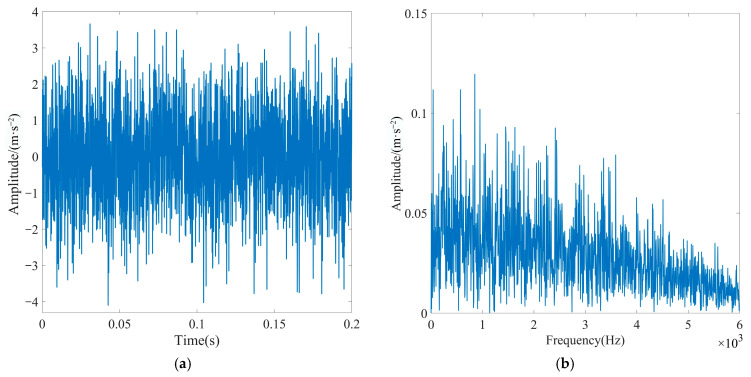
Basic characteristic diagram of the simulated signal: (**a**) time-domain diagram and (**b**) Hilbert envelope spectrogram.

**Figure 9 sensors-23-08620-f009:**
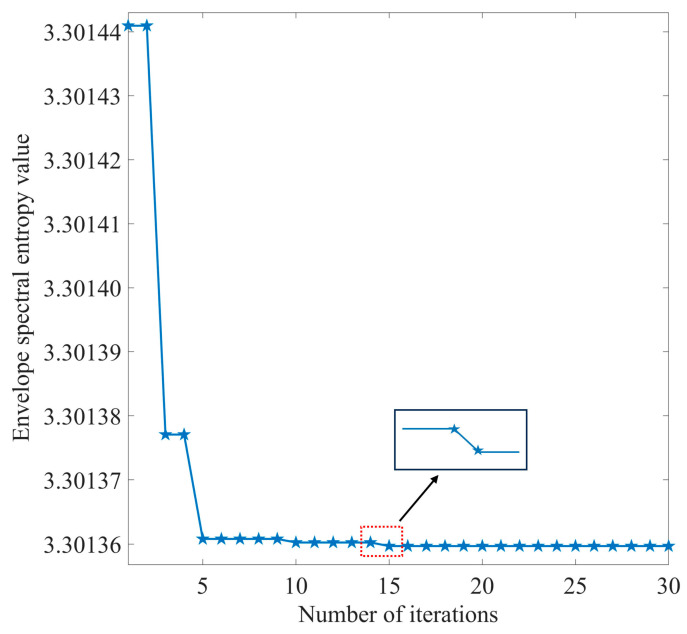
IPOA-VMD optimization iteration curve (simulated signal).

**Figure 10 sensors-23-08620-f010:**
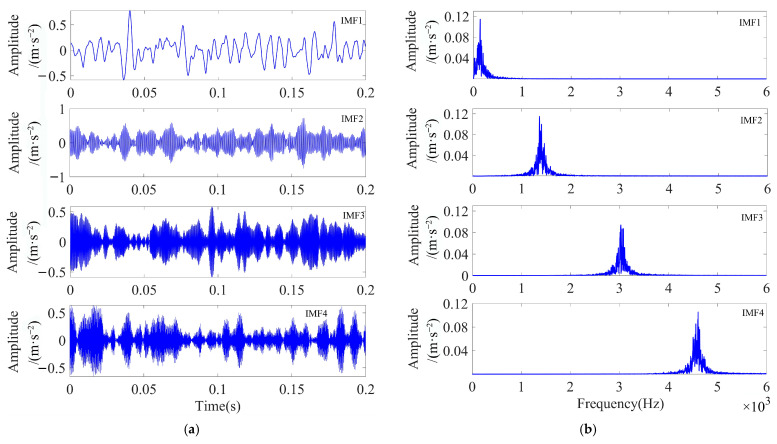
Time-frequency domain diagram of the IMF component after IPOA-VMD decomposition (simulated signal): (**a**) time domain and (**b**) frequency domain.

**Figure 11 sensors-23-08620-f011:**
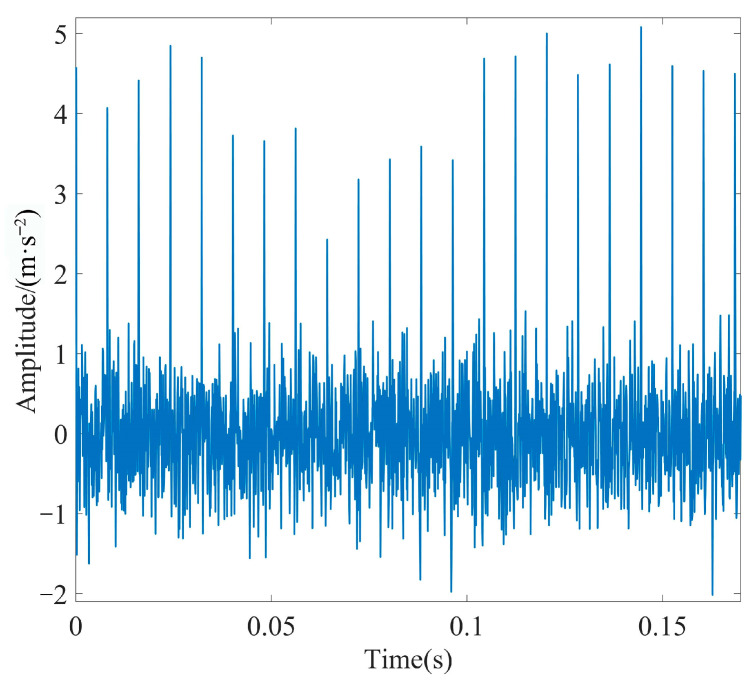
Time-domain diagram of the processed IMF1 using MOMEDA.

**Figure 12 sensors-23-08620-f012:**
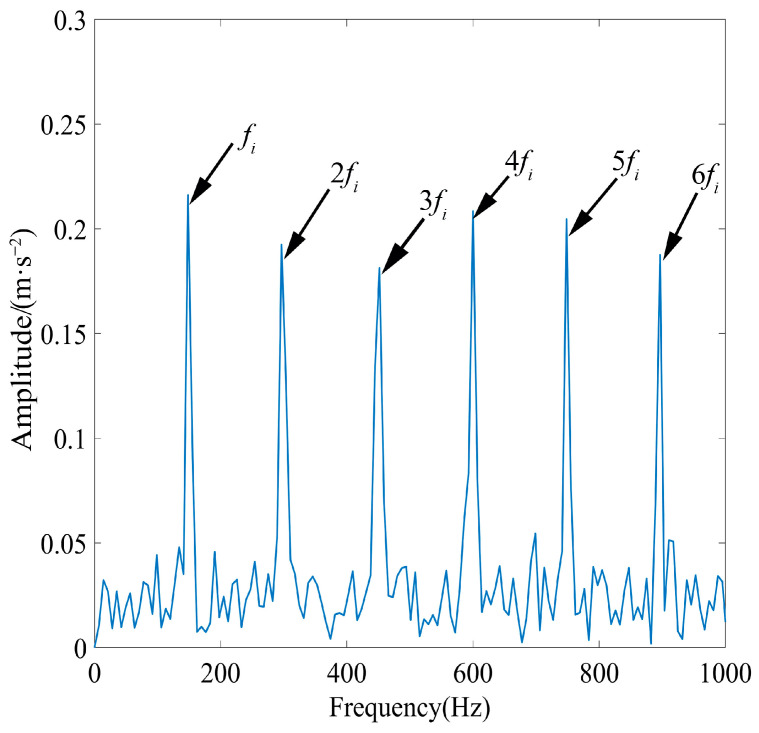
TEO spectrum of IMF1 after MOMEDA processing.

**Figure 13 sensors-23-08620-f013:**
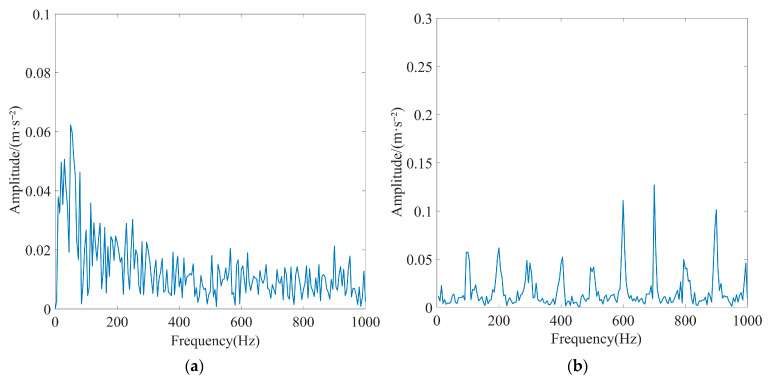
Results of IMF1 processing by different methods: (**a**) TEO demodulation analysis and (**b**) MOMEDA processing.

**Figure 14 sensors-23-08620-f014:**
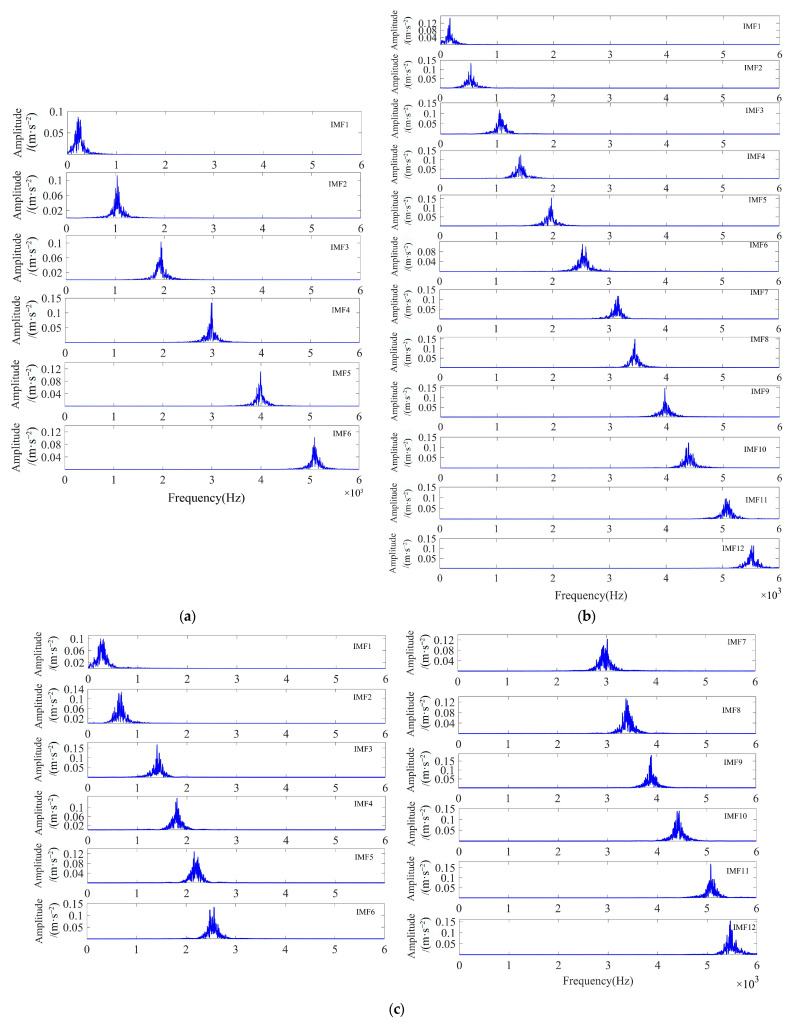
Spectrum of each component of simulated signals with different SNRs after IPOA-VMD decomposition: (**a**) SNR equals −16 dB, (**b**) SNR equals −17 dB, and (**c**) SNR equals −18 dB.

**Figure 15 sensors-23-08620-f015:**
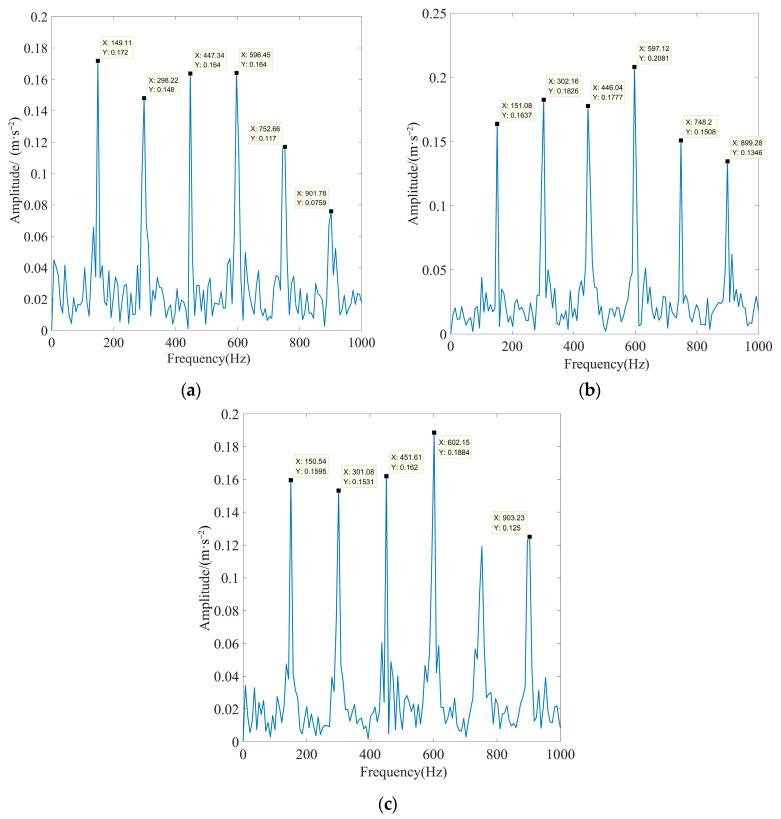
TEO spectrum of optimal IMF of simulated signals with different SNRs after MOMEDA processing: (**a**) SNR equals −16 dB, (**b**) SNR equals −17 dB, and (**c**) SNR equals −18 dB.

**Figure 16 sensors-23-08620-f016:**
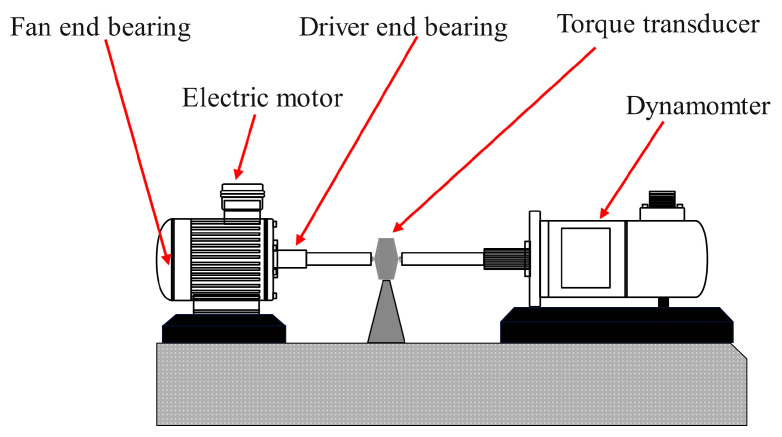
Schematic of the rolling bearing fault experimental platform.

**Figure 17 sensors-23-08620-f017:**
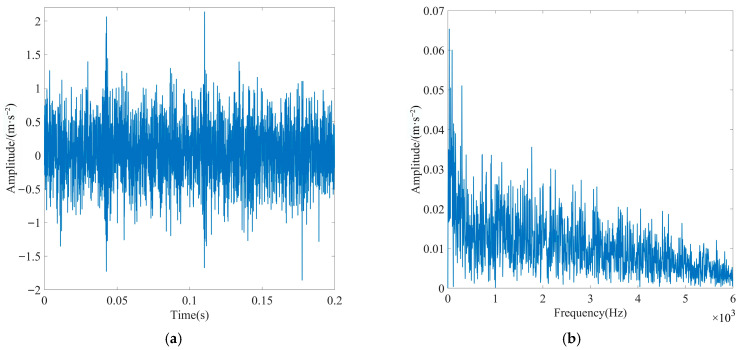
Basic characteristic diagram of inner ring fault signals: (**a**) time domain and (**b**) Hilbert envelope spectrum.

**Figure 18 sensors-23-08620-f018:**
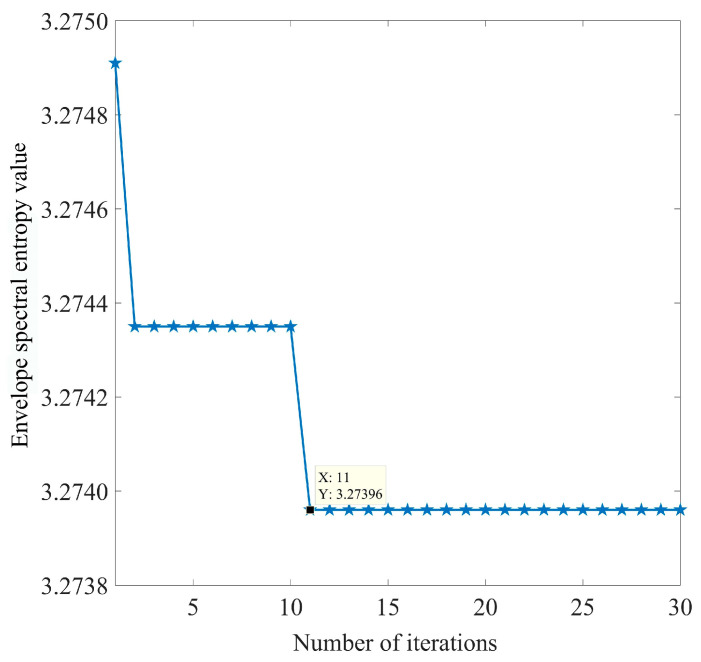
IPOA-VMD optimization search iteration curve (actual signal).

**Figure 19 sensors-23-08620-f019:**
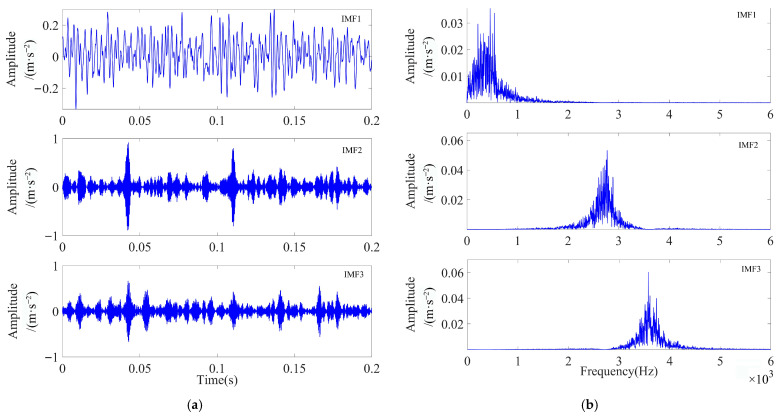
Time- and frequency-domain waveforms of each IMF component after IPOA-VMD decomposition (actual signal): (**a**) time domain and (**b**) frequency domain.

**Figure 20 sensors-23-08620-f020:**
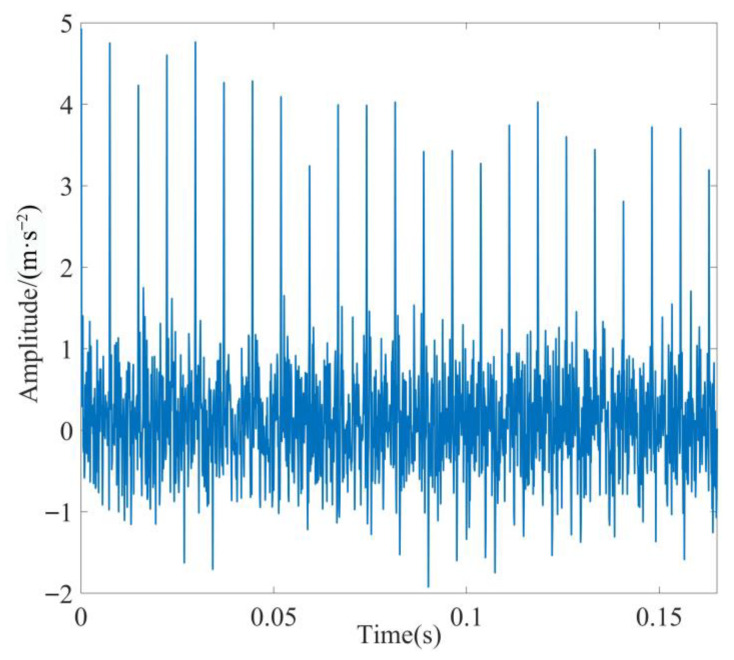
Time-domain diagram of IMF2 processed by MOMEDA.

**Figure 21 sensors-23-08620-f021:**
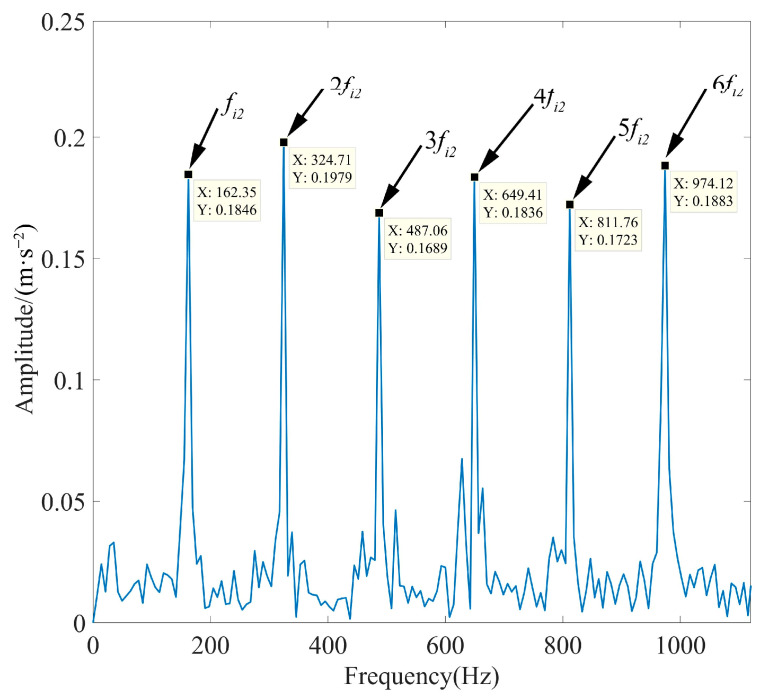
TEO spectrum of IMF2 after MOMEDA processing.

**Figure 22 sensors-23-08620-f022:**
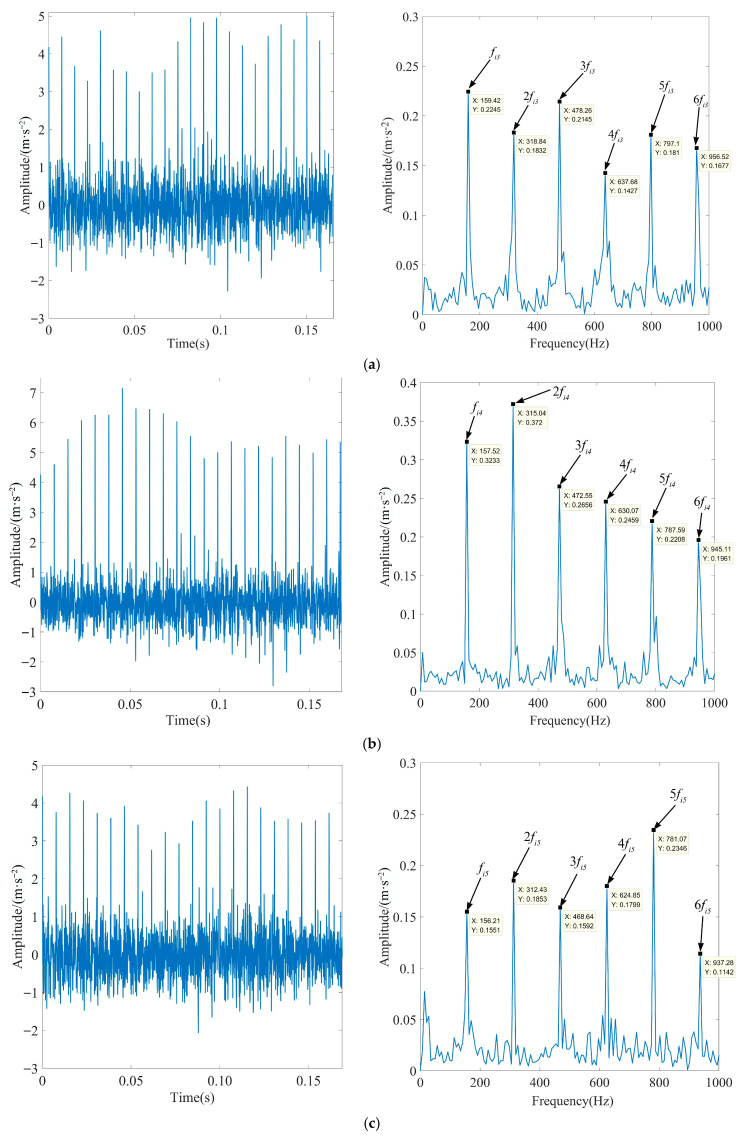
Time-domain waveforms and TEO spectrum of each optimal IMF of the fault signal under different loads after processing via MOMEDA: (**a**) motor load is 50%, (**b**) motor load is 100%, and (**c**) motor load is 150%.

**Figure 23 sensors-23-08620-f023:**
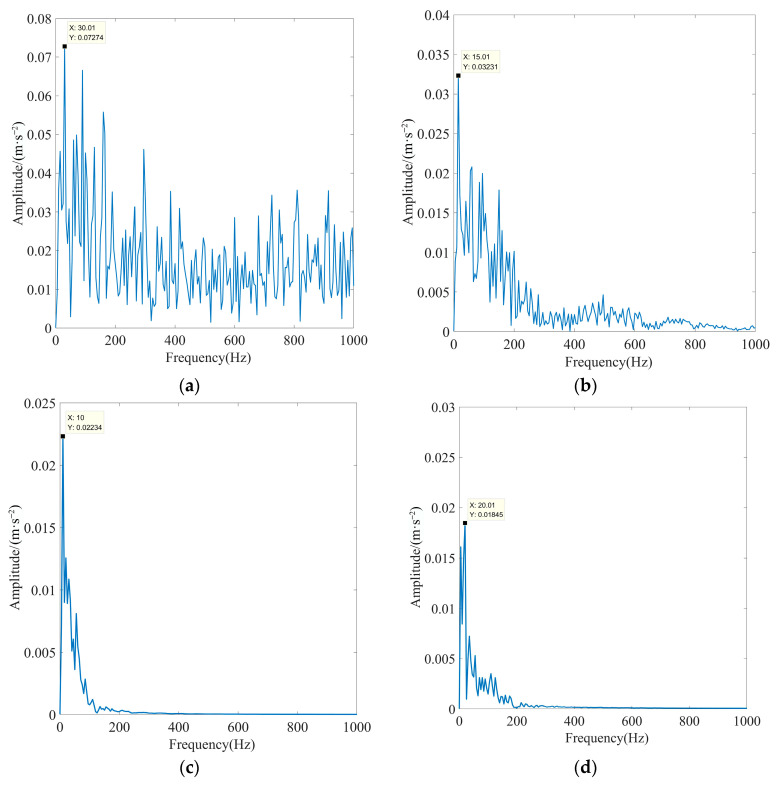
Optimal IMF envelope spectra of rolling bearing inner race fault signals obtained using the EEMD algorithm under different loads: (**a**) motor load is 0%, (**b**) motor load is 50%, (**c**) motor load is 100%, and (**d**) motor load is 150%.

**Figure 24 sensors-23-08620-f024:**
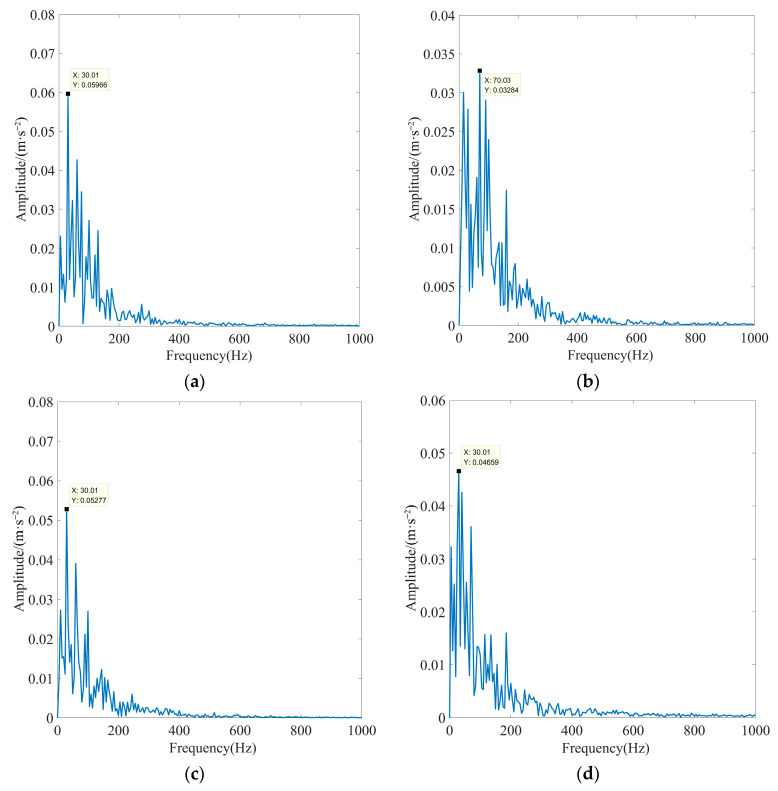
Optimal IMF envelope spectra of rolling bearing inner race fault signals obtained using the FP-VMD algorithm under different loads: (**a**) motor load is 0%, (**b**) motor load is 50%, (**c**) motor load is 100%, and (**d**) motor load is 150%.

**Figure 25 sensors-23-08620-f025:**
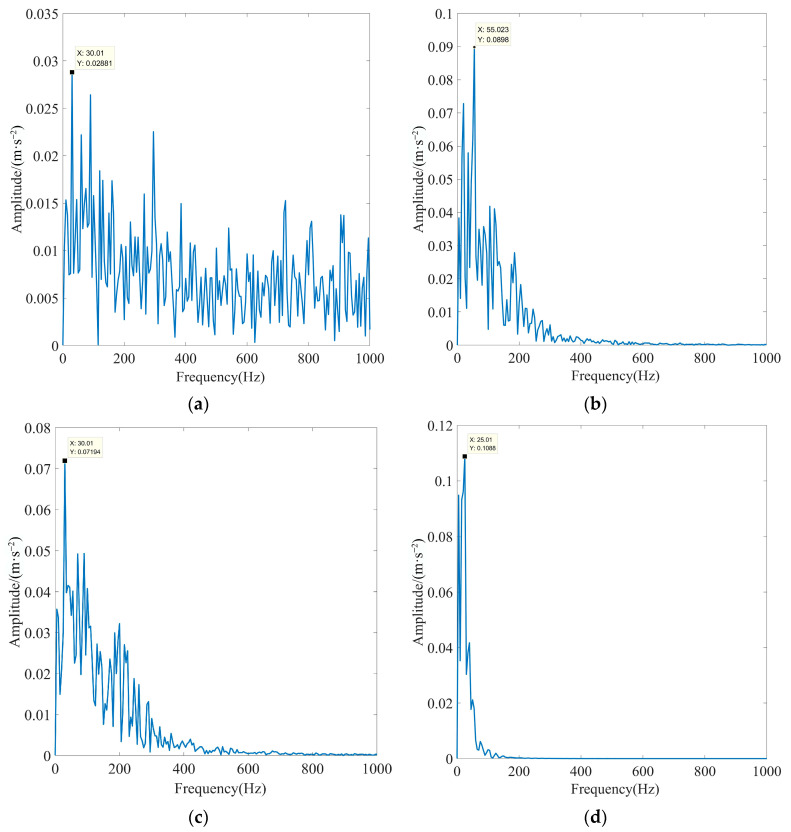
Optimal IMF envelope spectra of rolling bearing inner race fault signals obtained using the EWT-ICA algorithm under different loads: (**a**) motor load is 0%, (**b**) motor load is 50%, (**c**) motor load is 100%, and (**d**) motor load is 150%.

**Table 1 sensors-23-08620-t001:** Benchmarking function parameters.

Benchmark Function Expression	Dimension	Value Range	Optimal Solution
$F_{1}={\sum_{i=1}^{n}\left(\sum_{j=1}^{i}x_{j}\right)}^{2}$	30	[−100,100]	0
$F_{2}=\underset{i}{max}\left\{\left|\left.x_{i}\right|,1\leq i\leq n\right.\right\}$	30	[−100,100]	0
$F_{3}=\sum_{i=1}^{n}\left[x_{i}^{2}-10cos(2\pi x_{i})+10\right]$	30	[−5.12,5.12]	0
$F_{4}=-20exp\left(-0.2\sqrt{\frac{1}{n}\sum_{i=1}^{n}x_{i}^{2}}\right)+20+e-exp\left(\frac{1}{n}\sum_{i=1}^{n}cos\left(2\pi x_{i}\right)\right)$	30	[−32,32]	0

**Table 2 sensors-23-08620-t002:** Comparison of the test results of different optimization algorithms.

Function	Statistic	DS	GOA	MPA	PSO	AOA	POA	IPOA
*F* _1_	Avg	5.9020 × 10^4^	5.2200 × 10^3^	1.5787 × 10^2^	1.5218 × 10^3^	1.0680 × 10^−10^	5.5570 × 10^−17^	0
	MR	0	0	0	0	53	51	97
*F* _2_	Avg	5.9990 × 10^1^	1.9120 × 10^1^	3.9619 × 10^−1^	5.4721 × 10^0^	7.8481 × 10^−7^	1.0567 × 10^−10^	0
	MR	0	0	0	0	21	32	97
*F* _3_	Avg	1.9642 × 10^2^	1.4376 × 10^2^	1.7388 × 10^1^	2.8763 × 10^2^	3.4106 × 10^−13^	0	0
	MR	0	0	0	0	61	49	98
*F* _4_	Avg	9.7789 × 10^0^	1.1971 × 10^1^	8.5541 × 10^−2^	5.6286 × 10^0^	1.9966 × 10^1^	1.1888 × 10^−10^	8.8817 × 10^−16^
	MR	0	0	0	0	0	28	97

**Table 3 sensors-23-08620-t003:** IPOA initialization parameters (simulated signal).

Parameter Name	Specific Value
population size	80
spatial dimension	2
number of iterations	30
penalty factor range	[100, 20,000]
range of values of decomposition layers	[2, 15]

**Table 4 sensors-23-08620-t004:** Evaluation parameters of each IMF component (simulated signal).

Index	IMF1	IMF2	IMF3	IMF4
*Kurtosis*	3.9309	2.7432	2.6481	3.3938
*SEGI*	0.5552	0.4590	0.4551	0.5636
*K-SEGI*	3.7923	1.0787	1.0000	2.9924

**Table 5 sensors-23-08620-t005:** Rolling bearing dataset.

Fault Condition	Fault Diameter(mm)	Motor Load (%)	Motor Speed (r/min)
Inner race fault	0.3556	0	1797
		50	1772
		100	1750
		150	1730

**Table 6 sensors-23-08620-t006:** Fundamental characteristics of the rolling bearing.

Bearing Model	Rolling Diameter	Pitch Diameter	Number of Balls	Contact Angle
SKF6205-2RS	7.94 mm	39.04 mm	9	0°

**Table 7 sensors-23-08620-t007:** IPOA initialization parameters.

Parameter Name	Specific Value
population size	80
spatial dimension	2
number of iterations	30
penalty factor range	[100, 20,000]
range of values of decomposition layers	[2, 15]

**Table 8 sensors-23-08620-t008:** Evaluation parameters of each IMF component (actual signal).

Index	IMF1	IMF2	IMF3
*Kurtosis*	2.8668	9.0051	5.3069
*SEGI*	0.4953	0.6656	0.6294
*K-SEGI*	1.0000	4.0000	2.2734

## Data Availability

The data were obtained from the website of Case Western Reverse Laboratory and is available at: https://engineering.case.edu/bearingdatacenter/download-data-file.

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
