# Peer review of "A Rolling Bearing Fault Feature Extraction Algorithm Based on IPOA-VMD and MOMEDA"

_sensors, 2023, doi:10.3390/s23208620_

Round 1
Reviewer 1 Report
The paper proposes a rolling bearing fault feature extraction algorithm based on IPOA-VMD and MOMEDA. The algorithm aims to accurately extract fault features from vibration signals captured by a sensor in the presence of strong background noise interference. The proposed method combines IPOA with VMD and MOMEDA to enhance the accuracy of fault feature extraction. Several major concerns and specific comments are listed below:
1. In the Introduction section, I noticed that the literature review is not comprehensive, and some of the cited literature seems to be outdated. I recommend that the authors expand the review to include more recent publications, particularly focusing on articles from this journal or related journals from the past three years. Doing so will ensure the relevance and up-to-dateness of the cited works.
2. I recommend the authors conduct and provide more detailed ablation studies to elucidate the contribution and significance of each component in their methodology.
3. The authors have correctly explained the motivation behind the fault feature extraction of rolling bearings, but they have not properly explained the motivation behind their research objective. Specifically, they did not clearly explain the rationale behind using the proposed method and their methodolo1gy.
4. Additionally, it would be beneficial if the authors carry out comparative experiments against existing feature extraction methods. This will not only validate the superiority of the proposed model but also provide readers with a clearer context of its performance relative to established techniques.
5. Only the most basic IPOA with VMD and MOMEDA models are used in this paper. The presented results seem trivial and the contribution is therefore questionable. Compared with recent deep learning based rolling bearing fault feature extraction approaches such as the studies in Energy, 10.1016/j.energy.2023.128442, and Measurement Science and Technology, 10.1088/1361-6501/acce55; authors need to do a better job of highlighting their contributions.
6. What are the limitations of the method proposed in this paper? Is there potential for future enhancements or refinements to address these limitations?
7. Lastly, I believe the 'Conclusion' section should be rewritten to better encapsulate the main findings and implications of the study.
The language of the paper could benefit from further refinement to enhance clarity and ease of understanding for readers. Additionally, it's important to ensure adherence to the journal's formatting guidelines. For instance, the first letter in the Section 8 should be capitalized."
Author Response
Dear reviewer,
We have revised the paper according to your comments. Please refer to the attachment for details.

Reviewer 2 Report
This is a nicely written paper. The paper presents a novel algorithm designed to isolate fault features from rolling bearing fault signals, typically masked by ambient noise. This algorithm enhances the Pelican Optimization Algorithm (POA) through a reverse learning approach, and applies the enhanced POA (IPOA) for variable modal decomposition (VMD) and multipoint optimal minimum entropy deconvolution adjustment (MOMEDA). The selection of optimal modal components is guided by the kurtosis-square envelope Gini coefficient criterion, which are subsequently processed by MOMEDA to yield the optimal deconvolution signal. The Teager Energy Operator (TEO) is employed to demodulate and scrutinize the signal, thereby amplifying the transient shock component of the original fault signal. The effectiveness of this method was confirmed through tests on both simulated and real signals, demonstrating its ability to accurately isolate failure traits despite significant interference from background noise.
Author Response
Dear reviewer
Thank you very much for taking the time to review this manuscript. Your comments are very helpful to improve the quality of our paper increase our confidence in our work. We want to thank you again from the bottom of our hearts for your comments.

Reviewer 3 Report
There are two points that the reviewer wants to address:
1. In general, the vibration signal analysis has lots of disadvantages such as, it is required accurate location to place the accelerometer sensors, these sensors are not easy to place especially for unaccessible machines, vibration signals are easily corrupted by noises from other machines vibrations (laboratory environment generally has much less noises compared to industrial environment), and it is not trivial to build continuous condition monitoring system based on vibration analysis as the sensors must be physically placed on the machines. Implementing such complex algorithms based on vibration analysis adds more difficulties to practically implement this for industrial practices.
2. The authors presented dataset with only one operation point. This condition can only happen in the laboratory environment and not in the industrial environment. The machines are operated in various operation points with a very limited length of time of a stable measurement. Obtaining very long stable vibration measurements is challenging.
Therefore, based on those two points, the reviewer cannot see any potential contributions to the practical industrial implementation of this topic.
However, this topic can still contribute to the practice of implementing advance signal processing algorithms for fault diagnosis applications.
There are some points that the authors have to clarify before the paper can be published in this journal such as,
1. There is lack of comparisons between proposed approach and mentioned previously research in reference [16-19]. Why is your algorithm better than those? What is the point of you propose this as there were already similar signal processing algorithms based on VMD? In order to find the answer on those questions you need to clarify it by making comparisons with those previously mentioned research.
2. There is lack of justification on selecting the optimization parameters mentioned in Table 2 and 5. Why did the range of penalty factors different between simulation and experiment?
3. What is the minimum length of time of the vibration signals for this approach to provide satisfactory results? As it is mentioned before that in practice industrial machines are operated in various operation points with a very limited length of time of a stable measurement.
4. What is the level of signal-to-noise ratio (SNR) presented in the simulation and in the experiment?Comparison of different SNR levels is needed to find out the limit of the proposed signal processing algorithm.
5. In the experiment section, since the IMF2 was selected for further processing with TEO-MOMEDA and seeing from the level of amplitude, it was expected that the signatures should be located in the spectrum bands of 2 - 4 kHz. However, in Fig 19, the signatures were located in bands less than 1 kHz with clear innering fault signatures with amplitude much higher than the IMF2 spectra. Can you provide further explanation on this?
6. Implementing only one dataset with one operation point does not enough to draw the conclusion bullet point 3. Because the vibration bearing signatures are located in different spectrum band locations as the severity level increases, and tend to be invisible when the machines running in low load (below 70%). With missing information of the load conditions, motor specification, and with the various operation points presented, it is really difficult to be convinced by the third conclusion drawn from this study.
Author Response
Dear reviewer:
Thank you very much for taking the time to review this manuscript. We have revised the paper according to your comments. Please refer to the attachment for details.

Round 2
Reviewer 1 Report
1. The extraction of fault features based on deep learning is the mainstream method currently. The authors should discuss this aspect in the introduction and demonstrate in the example section how their method is superior to the benchmark deep learning methods.
2. Justification is lacking for the use of IPOA-VMD and MOMEDA. Clarify immediately why these were chosen.
3. The authors have not adhered to the proper citation format for references. For instance, reference 23 should be cited as McDonald et al. [23]. Please correct the citation format throughout the manuscript.
4. The authors have failed to explicitly state their contributions as per the reviewer's request. Please rectify this oversight immediately.
5. The number of sections in this article is excessive, resulting in a disorganized and chaotic structure. Please streamline the structure for clarity and coherence.
6. Lastly, I recommend that the authors specifically showcase the modified portions in their response, to facilitate the reviewer’s reading and assessment.
Author Response
Dear reviewer:
Thank you very much for your comments, we have completed the revision according to your comments. For details, please refer to the attachment.

Reviewer 3 Report
Dear authors thank you for your effort in addressing the reviewer's comments. There are comments that have been addressed satisfactorily. However, there are still comments that have not been addressed satisfactorily. Hence, the reviewer feels that the paper is not ready for publication and must undergo another major revision.
Please address the following comments accordingly,
1. There is a lack of comparisons between the proposed approach and the mentioned previously research in reference [16-19]. What you have to do in order to answer this, is by making the results comparison between your proposed method and those mentioned methods. Then you can elaborate more of your paragraphs 3 and 4 as the discussion rather than in the introduction. Hence, you can convince the readers that your algorithm performs better than those.
2. Because you can adjust the parameters in Tables 2 and 6. The question is why in the first place do you select those values as your initial parameters? what is your justification for selecting those parameters? For example, Why is the population size set to be 80? why not other values? This applies to the other four described parameters.
3. I believe that you misunderstood the question "What is the minimum length of time of the vibration signals for this approach to provide satisfactory results?" Therefore, I would rephrase that into, what is the minimum data point in the time domain that you need to acquire for stable measurement? in order to obtain satisfactory results.
4. It is not common to describe motor load in kW, rather it is common to describe motor load in %. It leads to confusion because the motor runs at 0 kW means that the motor does not run at all. Hence, it is difficult for the reviewer to understand the operation condition described.
Author Response

(The authors gave the same response as above.)

Round 3
Reviewer 1 Report
The paper has significantly improved through revisions, and I recommend its acceptance.